# Effects of sustained inflation pressure during neonatal cardiopulmonary resuscitation of asphyxiated piglets

Gyu-Hong Shim [1,2☯], Seung Yeun Kim [1,3☯], Po-Yin Cheung[1,4], Tze-Fun Lee[1,4], Megan O'Reilly[1,4], Georg M. Schmölzer [1,4]*

1 Centre for the Studies of Asphyxia and Resuscitation, Neonatal Research Unit, Royal Alexandra Hospital, Edmonton, Alberta, Canada, 2 Department of Pediatrics, Inje University Sanggye Paik Hospital, Seoul, Korea, 3 Department of Pediatrics, Eulji University Hospital, Daejeon, Korea, 4 Department of Pediatrics, Faculty of Medicine and Dentistry, University of Alberta, Edmonton, Alberta, Canada

☯ These authors contributed equally to this work.
* georg.schmoelzer@me.com

**Data Availability Statement:** All relevant data are within the paper. Additional raw data not necessary to replicate the results can be requested from the corresponding author.

## Abstract

### Background

Sustained inflation (SI) during chest compression (CC = CC+SI) has been recently shown as an alternative method during cardiopulmonary resuscitation in neonates. However, the optimal peak inflation pressure (PIP) of SI during CC+SI to improve ROSC and hemodynamic recovery is unknown.

### Objective

To examine if different PIPs of SI during CC+SI will improve ROSC and hemodynamic recovery in severely asphyxiated piglets.

### Methods

Twenty-nine newborn piglets (1–3 days old) were anesthetized, intubated, instrumented and exposed to 30-min normocapnic hypoxia followed by asphyxia. Piglets were randomized into four groups: CC+SI with a PIP of 10 $cmH_2O$ (CC+SI_PIP_10, n = 8), a PIP of 20 $cmH_2O$ (CC+SI_PIP_20, n = 8), a PIP of 30 $cmH_2O$ (CC+SI_PIP_30, n = 8), and a sham-operated control group (n = 5). Heart rate, arterial blood pressure, carotid blood flow, cerebral oxygenation, and respiratory parameters were continuously recorded throughout the experiment.

### Results

Baseline parameters were similar between all groups. There was no difference in asphyxiation (duration and degree) between intervention groups. PIP correlated positively with tidal volume ($V_T$) and inversely with exhaled $CO_2$ during cardiopulmonary resuscitation. Time to ROSC and rate of ROSC were similar between piglets resuscitated with CC+SI_PIP_10, CC+SI_PIP_20, and CC+SI_PIP_30 $cmH_2O$: median (IQR) 75 (63–193) sec, 94 (78–210)

**Funding:** We would like to thank the public for donating money to our funding agencies: GMS is a recipient of the Heart and Stroke Foundation/ University of Alberta Professorship of Neonatal Resuscitation, a National New Investigator of the Heart and Stroke Foundation Canada and an Alberta New Investigator of the Heart and Stroke Foundation Alberta. The study was supported by a Grant from the SickKids Foundation in partnership with the Canadian Institutes of Health Research (CIHR - Institute of Human Development, Child and Youth Health (IHDCYH)), New Investigator Research Grant Program (Grant number - No. NI17-033). This research has been facilitated by the Women and Children's Health Research Institute through the generous support of the Stollery Children's Hospital Foundation.

**Competing interests:** The authors have declared that no competing interests exist.

sec, and 85 (70–90) sec; 5/8 (63%), 7/8 (88%), and 3/8 (38%) (p = 0.56 and p = 0.12, respectively). All piglets that achieved ROSC survived to four hours post-resuscitation. Piglets resuscitated with CC+SI_PIP_30 $cmH_2O$ exhibited increased concentrations of pro-inflammatory cytokines interleukin-1β and tumour necrosis factor-α in the frontoparietal cerebral cortex (both p<0.05 vs. sham-operated controls).

## Conclusion

In asphyxiated term newborn piglets resuscitated by CC+SI, the use of different PIPs resulted in similar time to ROSC, but PIP at 30 $cmH_2O$ showed a larger $V_T$ delivery, lower exhaled $CO_2$ and increased tissue inflammatory markers in the brain.

## Introduction

An estimated 10% of newborns need assistance to establish effective ventilation at birth, which remains the most critical step of neonatal resuscitation [1]. The need for chest compressions (CC) in the delivery room is rare (approximately 0.1% of term infants and up to 15% of preterm infants) [2–8]. A systemic review of newborns born between 1991 and 2004 who underwent prolonged chest compressions without signs of life at 10 minutes following birth noted 83% mortality, with 94% of survivors suffering death or severe disability [9]. Since the prognosis associated with receiving CC alone or epinephrine in the delivery room is poor, there is question whether the outcome can be improved with cardiopulmonary resuscitation methods specifically tailored to newborns [10, 11].

Current neonatal resuscitation guidelines recommend using an initial peak inspiratory pressure (PIP) of 20 to 25 $cmH_2O$ if positive pressure ventilation (PPV) is required during neonatal resuscitation; this can potentially be increased to a PIP of 30 to 40 $cmH_2O$ in some term infants [12–14]. There is evidence that a PIP of 20 to 25 $cmH_2O$ causes high tidal volume ($V_T$) delivery in preterm infants [15, 16], and excessive $V_T$ delivery induced by high PIP may cause hypocarbia [17], which is associated with brain injury [18]. Recently, Mian *et al* observed that preterm infants <29 weeks' gestation who received mask PPV with $V_T$>6 mL/kg have significantly higher rates of intraventricular hemorrhage [19]. However, to date no study has measured the optimal PIP and the adequate $V_T$ delivery during CC in the delivery room.

Despite extensive research over the past decade, the most effective way to deliver CC remains controversial. We have previously shown that providing a sustained inflation (SI) during CC (CC+SI) can significantly improve return of spontaneous circulation (ROSC) and survival in newborn piglets [20–23]. Our primary aim was to determine whether different PIP during CC+SI would affect time to ROSC and survival in newborn piglets with asphyxia-induced asystole. We hypothesized that using CC+SI with a PIP of 10 or 20 $cmH_2O$ would reduce the time needed to achieve ROSC compare to using a PIP of 30 $cmH_2O$. The secondary aim was to examine the effect of PIP on hemodynamics and respiratory parameters and brain inflammation after resuscitation.

## Materials and methods

Twenty-nine newborn mixed breed piglets were obtained on the day of experimentation from the University Swine Research Technology Centre. All experiments were conducted in accordance with the guidelines and approval of the Animal Care and Use Committee (Health

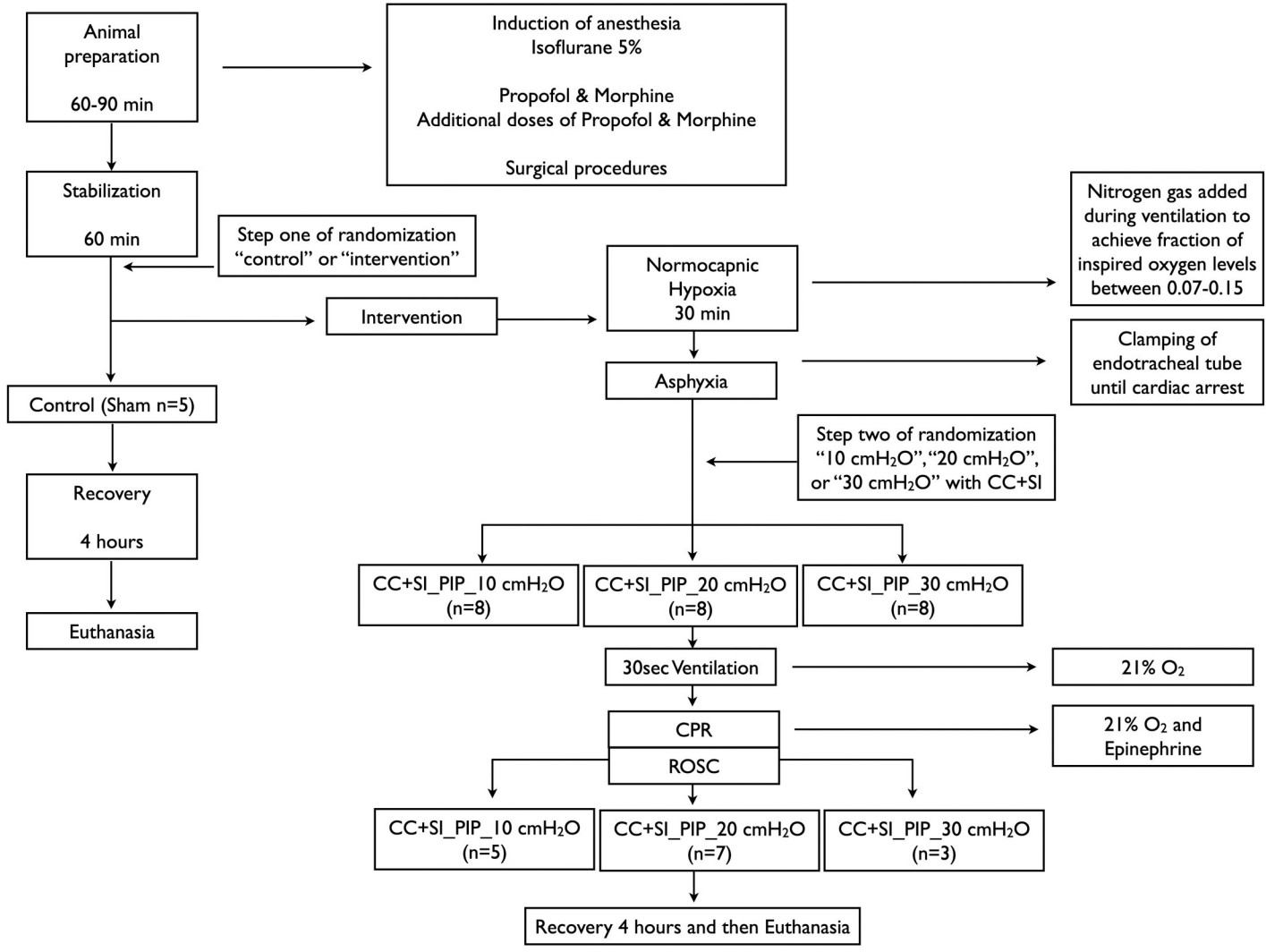

**Fig 1. Study flow chart.** CC+SI, chest compressions during sustained inflation; CPR, cardiopulmonary resuscitation.

Sciences), University of Alberta AUP00002151, according to the ARRIVE guidelines [24], and registered at preclinicaltrials.eu (PCTE0000149). The study protocol is presented in Fig 1.

## Randomization

Piglets were randomly allocated to control (sham-operated) or intervention (CC+SI with PIP of 10, 20, or 30 cmH₂O) groups. Allocation was block randomized with variable sized blocks using a computer-generated randomization program (http://www.randomizer.org). Sequentially numbered, sealed, brown envelopes containing the allocation were opened during the experiment (Fig 1).

## Sample size and power estimates

Our primary outcome measure was time to achieve ROSC. Our previous studies showed a mean ± SD ROSC of 120±25sec during resuscitation using CC+SI with a PIP of 30 cmH₂O. To test the primary hypothesis that CC+SI with a PIP of 10 cmH₂O or 20 cmH₂O would reduce

time to achieve ROSC, a sample size of 24 piglets (8 per group) would be was sufficient to detect a clinically important (30%) reduction in time to achieve ROSC (i.e. 84sec vs. 120sec), with 80% power and a 2-tailed alpha error of 0.05.

## Animal preparation

Piglets were instrumented as previously described with modifications [20, 25, 26]. Following the induction of anaesthesia using isoflurane, piglets were intubated via a tracheostomy, and pressure-controlled ventilation (Acutronic Fabian HFO; Hirzel, Switzerland) was commenced at a respiratory rate of 16–20 breaths/min and pressure of 20/5 cmH$_2$O. Oxygen saturation was kept within 90–100%, glucose level and hydration was maintained with an intravenous infusion of 5% dextrose at 10 mL/kg/hr. Throughout the entire experiment period, anaesthesia was maintained with intravenous propofol 5–10 mg/kg/hr and morphine 0.1 mg/kg/hr. Additional doses of propofol (1–2 mg/kg) and morphine (0.05–0.1 mg/kg) were also given as needed. The piglet's body temperature was maintained at 38.5–39.5˚C using an overhead warmer and a heating pad.

## Hemodynamic parameters

A 5-French Argyle® (Klein-Baker Medical Inc. San Antonio, TX) double-lumen catheter was inserted via the right femoral vein for administration of fluids and medications. A 5-French Argyle® single-lumen catheter was inserted above the right renal artery via the femoral artery for continuous arterial blood pressure monitoring in addition to arterial blood gas measurements. The right common carotid artery was also exposed and encircled with a real-time ultrasonic flow probe (2mm; Transonic Systems Inc., Ithaca, NY) to measure cerebral blood flow.

Piglets were placed in supine position and allowed to recover from surgical instrumentation until baseline hemodynamic measures were stable (minimum of one hour). Ventilator rate was adjusted to keep the partial arterial CO$_2$ between 35–45 mmHg as determined by periodic arterial blood gas analysis. Mean systemic arterial pressure, systemic systolic arterial pressure, heart rate, and percutaneous oxygen saturation were continuously measured and recorded throughout the experiment with a Hewlett Packard 78833B monitor (Hewlett Packard Co., Palo Alto, CA).

## Respiratory parameters

A respiratory function monitor (NM3, Respironics, Philips, Andover, MA) was used to continuously measure V$_T$, airway pressures, gas flow, and exhaled CO$_2$ (ETCO$_2$). The combined gas flow and ETCO$_2$ sensor was placed between the endotracheal tube and the ventilation device. V$_T$ was calculated by integrating the flow signal [27]. ETCO$_2$ was measured using non-dispersive infrared absorption technique. The accuracy for gas flow is ±0.125 L/min, ETCO$_2$ ±2 mmHg [28].

## Cerebral perfusion

Cerebral oxygenation (crSO$_2$) was measured using the Invos™ Cerebral/Somatic Oximeter Monitor (Invos 5100, Somanetics Corp., Troy, MI). The sensors were placed on the right forehead of the piglet and secured with wrap and tape. Light shielding was achieved with a slim cap. The Invos™ Cerebral/Somatic Oximeter Monitor calculates crSO$_2$, which is expressed as the percentage of oxygenated haemoglobin (oxygenated haemoglobin/total haemoglobin). Values of regional oxygen saturation were stored every second with a sample rate of 0.13 Hz [29].

## Experimental protocol

Piglets were randomized into four groups: CC+SI with a PIP of 10 cmH$_2$O (CC+SI_PIP_10, n = 8), CC+SI with a PIP of 20 cmH$_2$O (CC+SI_PIP_20, n = 8), CC+SI with a PIP of 30 cmH$_2$O (CC+SI_PIP_30, n = 8), or sham-operated controls (n = 5). To reduce selection bias, a two-step randomization process was used. Following surgical instrumentation and stabilization procedure, a subsequently numbered, sealed brown envelope containing the assignment "control" or "intervention" was opened (step one) (Fig 1). Piglets randomized to "intervention" underwent both hypoxia and asphyxia, whereas, the piglets randomized to "control" did not. The sham-operated control group received the same surgical protocol, stabilization, and equivalent experimental periods without hypoxia and asphyxia. The piglets that were randomized to "intervention" were exposed to 30 minutes of normocapnic hypoxia, which was followed by asphyxia. Asphyxia was achieved by disconnecting the ventilator and clamping the endotracheal tube until asystole. Asystole was defined as no heart rate audible during auscultation with standard stethoscope by a single investigator (GMS), who was blinded to HR displayed by ECG and carotid blood flow. After asystole was diagnosed a second subsequently numbered, sealed brown envelope containing the assignment "CC+SI_PIP_10 cmH$_2$O", "CC+SI_PIP_20 cmH$_2$O", or "CC+SI_PIP_30 cmH$_2$O" was opened (step two) (Fig 1). Fifteen seconds after asystole was diagnosed PPV was performed for 30 seconds with a Neopuff T-Piece (Fisher & Paykel, Auckland, New Zealand). The default settings of the experiment were a PIP according to group allocation (CC+SI_PIP_10, 20, or 30 cmH$_2$O), a positive end expiratory pressure of 5 cmH$_2$O, and a gas flow of 8 L/min using a fraction of inspired oxygen of 0.21 (Fig 1). Using the two-thumb hand-encircling technique [12–14], CC was performed at a rate of 90/min using a metronome by a single operator in all piglets. After 30 seconds of PPV, CC was started and SI was given for 20 seconds, and then paused for 1 second before resuming another SI of 20 seconds. This was repeated until the piglet achieved ROSC. Epinephrine (0.02 mg/kg per dose) was administered intravenously 2 minutes after the start of PPV, and administered every 3 minutes as needed if no ROSC was observed, to a maximum of four doses. ROSC was defined as an unassisted heart rate ≥100 bpm for 15 seconds. After ROSC, piglets recovered for four hours before being euthanized with an intravenous overdose of sodium pentobarbital (120 mg/kg).

## Data collection and analysis

Demographics of study piglets were recorded. Transonic flow probes, heart rate and pressure transducer outputs were digitized and recorded with LabChart® programming software (ADInstruments, Houston, TX). Airway pressures, gas flow, V$_T$, and ETCO$_2$ were measured and analyzed using Flow Tool Physiologic Waveform Viewer (Philips Healthcare, Wallingford, CT, USA). Following euthanization, the brain was removed from the skull and placed in ice-cold 2-methylbutane for 10 minutes before being stored at -80˚C. Tissue samples were only collected from piglets that survived four hours after the resuscitation. The frontoparietal cortex was isolated from the whole brain and was homogenized in phosphate buffer (50mM containing 1mM EDTA, pH 7.0). Homogenates were centrifuged (3,000x$g$ for 10min at 4˚C), the supernatants were collected, and protein concentration was quantified using the Bradford method. The concentrations of pro-inflammatory cytokines interleukin (IL)-1β, IL-6, and tumour necrosis factor (TNF)-α in brain tissue homogenates were determined using commercially available ELISA kits (PLB00B, P6000B, PTA00; R&D Systems, Minneapolis, USA). Cytokine concentrations were expressed relative to protein concentrations. The data are presented as mean (SD) for normally distributed continuous variables and median (IQR) when the distribution was skewed. For all respiratory parameters, continuous values during resuscitation

were analyzed. The data was tested for normality and compared using 2-way ANOVA for repeated measures using Bonferoni post-test. *P*-values are 2-sided and p<0.05 was considered statistically significant. Statistical analyses were performed with SigmaPlot (Systat Software Inc., San Jose, USA).

## Results

Twenty-nine newborn mixed breed piglets 1–3 days old, weighing 2.0 (0.13)kg, were obtained on the day of the experiment and were randomly assigned to either CC+SI_PIP_10, CC+-SI_PIP_20, CC+SI_PIP_30, or sham-operated group. There were no significant differences in the baseline parameters between groups (Table 1).

### Resuscitation

Although there were no significant differences in asphyxia time between groups, there was a trend for a shorter asphyxia time in the CC+SI_PIP_10 group (Table 2). However, this did not result in a difference in the degree of asphyxiation (as indicated by pH, $PaCO_2$, base excess, and lactate) between the intervention groups (Table 2). During resuscitation, the proportion of piglets that received epinephrine was similar between groups: 5/8 (63%) piglets in the CC+-SI_PIP_10 group, 4/8 (50%) piglets in the CC+SI_PIP_20 group, and 5/8 (63%) piglets in the CC+SI_PIP_30 group (p = 0.84). Overall, 7/8 (88%) piglets in the CC+SI_PIP_20 group survived, compared to 5/8 (63%) piglets in the CC+SI_PIP_10 group and 3/8 (38%) piglets in the CC+SI_PIP_30 group (p = 0.12). Resuscitation time to achieve ROSC was similar between groups (Table 2). All piglets that achieved ROSC survived for four hours after resuscitation (Table 2).

### Respiratory parameters

Respiratory parameters are presented in Table 3. As expected, there were significant differences in the respiratory parameters between groups, due to the use of different PIP to deliver the SI. The $V_T$ increased as the PIP used to deliver SI increased: CC+SI_PIP_10: 7.3 (3.3) mL/kg; CC+SI_PIP_20: 10.3 (3.1) mL/kg; CC+SI_PIP_30: 14.0 (3.3) mL/kg; (p = 0.0018). Similarly, 30 cmH$_2$O PIP resulted in lower exhaled $CO_2$ compared to 10 and 20 cmH$_2$O (CC+SI_PIP_30:

**Table 1. Baseline characteristics.**

|  | Sham-operated (n = 5) | CC+SI_PIP_10 (n = 8) | CC+SI_PIP_20 (n = 8) | CC+SI_PIP_ 30 (n = 8) |
|---|---|---|---|---|
| Age (days) | 2.0 (1–2.5) | 2.5 (1–3) | 2.0 (1–2) | 2.0 (1–3) |
| Weight (kg) | 2.0 (1.8–2.2) | 2.0 (1.8–2.0) | 2.1 (2.0–2.1) | 2.1 (1.9–2.2) |
| Heart rate (bpm) | 198 (165–220) | 196 (188–249) | 209 (176–221) | 176 (171–212) |
| Mean Arterial blood pressure (mmHg) | 67 (60–72) | 60 (52–69) | 54 (49–59) | 56 (53–58) |
| Carotid flow (mL/min/kg) | 49 (30–57) | 46 (31–49) | 41 (30–54) | 42 (38–51) |
| Cerebral oxygenation (%) | 59 (54–64) | 56 (50–63) | 55 (48–62) | 53 (46–60) |
| pH | 7.48 (7.46–7.55) | 7.44 (7.37–7.52) | 7.52 (7.47–7.55) | 7.50 (7.45–7.52) |
| $PaCO_2$ (torr) | 31.4 (28.1–34.9) | 35.3 (25.1–38.3) | 31.3 (28.0–36.6) | 33.7 (31.1–34.4) |
| $SpO_2$ (%) | 98 (98–99) | 98 (97–99) | 98 (96–99) | 99 (97–99) |
| Base excess (mmol/L) | 2.0 (1–5) | 0.5 (-4.8–1.8) | 4.0 (-1-5) | 2.5 (1.2–5.5) |
| Lactate (mmol/L) | 3.1 (3.0–3.9) | 4.2 (3.1–4.9) | 4.0 (2.9–5.3) | 3.7 (3.0–4.6) |

Data are presented as median (IQR)

**Table 2. Characteristics of asphyxia, resuscitation and survival of asphyxiated piglets.**

| | | CC+SI_PIP_10 (n = 8) | CC+SI_PIP_20 (n = 8) | CC+SI_PIP_30 (n = 8) | p-value |
|---|---|---|---|---|---|
| Asphyxia time (sec) | | 240 (156–415) | 522 (338–600) | 500 (364–557) | 0.06 |
| After asphyxiation | pH | 6.54 (6.50–6.75) | 6.51 (6.50–6.67) | 6.60 (6.50–6.72) | 0.66 |
| | $PaCO_2$ (torr) | 93 (68–123) | 98 (80–124) | 95 (86–105) | 0.91 |
| | BE (mmol/L) | -30 (-30~-27) | -30 (-30~-28) | -29 (-30~-23) | 0.55 |
| | Lactate (mmol/L) | 14 (12–16) | 16 (14–19) | 14 (12–17) | 0.28 |
| Resuscitation | Received epinephrine (n(%)) | 5 (63) | 4 (50) | 5 (63) | 0.84 |
| | Epinephrine doses (n) | 1.0 (0–4) | 0.5 (0–2) | 4 (0–4) | 0.28 |
| Achieving ROSC (n (%)) | | 5 (63) | 7 (88) | 3 (38) | 0.12 |
| ROSC time (sec) | | 75 (63–193) | 94 (78–210) | 85 (70–90) | 0.56 |
| Survival 4h after ROSC (n (%)) | | 5 (100) | 7 (100) | 3 (100) | 1.00 |

Data are presented as median (IQR); BE–base excess; ROSC–return of spontaneous circulation

10.8(4.5) mmHg vs. CC+SI_PIP_20: 16.7 (10.6) vs. CC+SI_PIP_10: 26.8(8.5) mmHg; (p = 0.0032) (Table 3).

## Changes in hemodynamic parameters

Hemodynamic changes of all groups are presented in Fig 2. At baseline, there was no significant difference in heart rate, mean arterial blood pressure, carotid blood flow, and cerebral oxygen saturation between all groups. At the end of asphyxia, all hemodynamic parameters were significantly reduced in the intervention groups compared to sham-operated controls. Following resuscitation and reoxygenation, heart rate returned to similar values as controls. Although mean arterial blood pressure increased towards baseline after resuscitation, values in the CC+SI_PIP_20 group remained lower than the baseline value at the end of the experiment. At the end of reoxygenation, the carotid blood flow was markedly reduced in CC+SI_PIP_20 and CC+SI_PIP_30 groups, resulting in lower cerebral oxygenation as compared with their own baseline values. However, there were no statistical differences between groups at the end of the reoxygenation period.

## Brain injury markers

The concentrations of IL-1β and TNF-α in frontoparietal cortex tissue were significantly greater in the CC+SI_PIP_30 group compared to sham-operated controls (Fig 3). There was no significant difference in IL-6 concentrations between all groups (Fig 3).

**Table 3. Respiratory parameters before ROSC.**

| | CC+SI_PIP_10 (n = 8) | CC+SI_PIP_20 (n = 8) | CC+SI_PIP_30 (n = 8) | p value |
|---|---|---|---|---|
| Peak Inflation Flow | 4.5 (1.2) | 6.7 (1.9) | 8.4 (1.9) | **0.0006** |
| Peak Expiration Flow | -7.1 (2.0) | -9.2 (2.8) | -12.2 (3.1) | **0.0039** |
| Peak Inflation Pressure (cm $H_2O$) | 15.7 (4.7) | 25.6 (1.2) | 34.3 (3.8) | **<0.0001** |
| Positive End Expiratory Pressure (cm $H_2O$) | 17.4 (5.6) | 26.5 (3.2) | 34.2 (5.4) | **<0.0001** |
| Exhaled $CO_2$ (mmHg) | 26.8 (8.5) | 16.7 (10.6) | 10.8 (4.5) | **0.0032** |
| Tidal Volume (mL/kg) | 7.3 (3.3) | 10.3 (3.1) | 14.0 (3.3) | **0.0018** |
| Rate (/min)* | 90 (1) | 90 (1) | 90 (1) | 1.000 |
| Minute Ventilation (mL/kg/min) | 657 (297) | 927 (279) | 1,260 (297) | **0.0018** |

Data are presented as mean (SD)

*Rate = Ventilation Rate = number of CC corresponds with number of ventilations per min

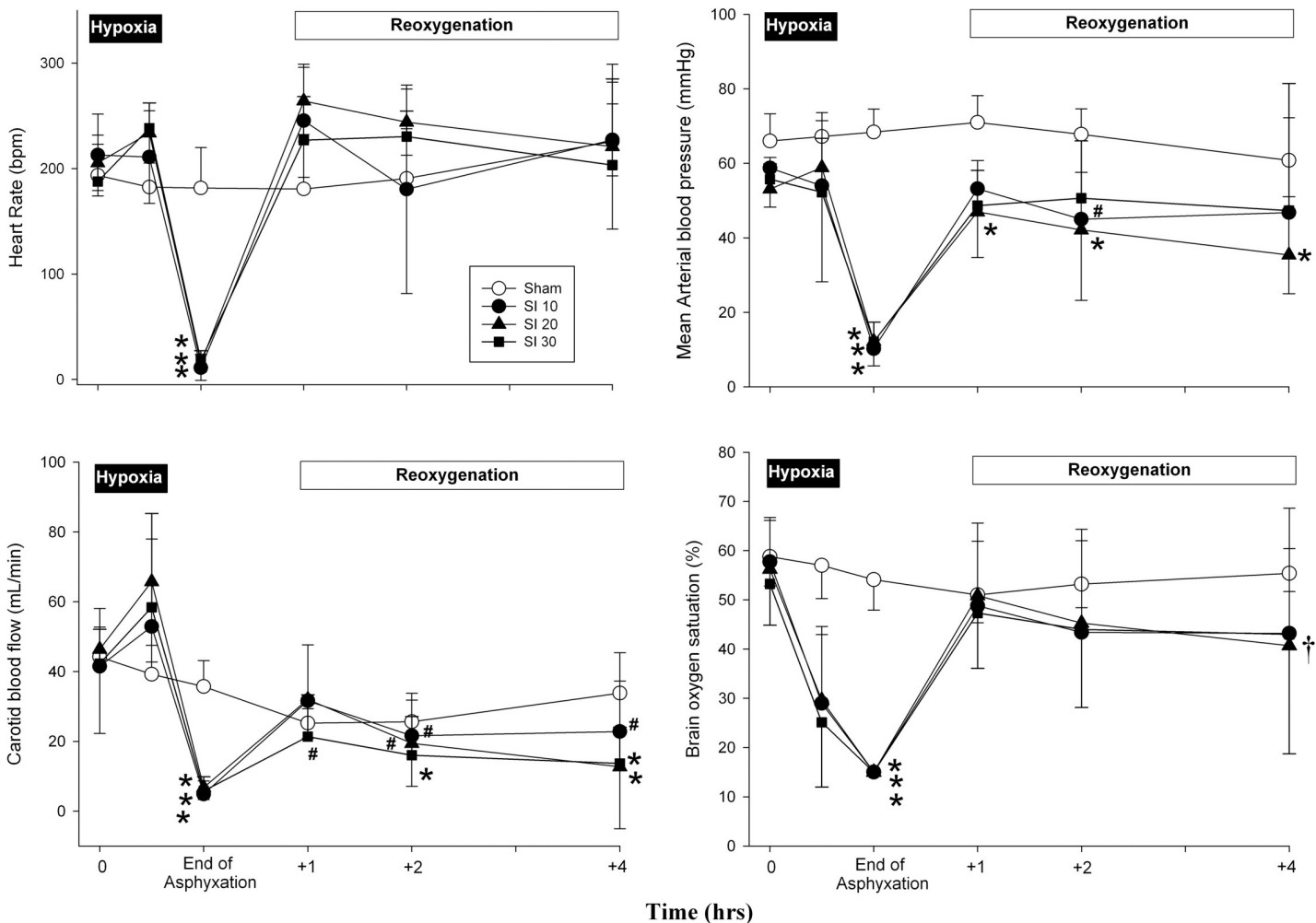

**Fig 2.** Hemodynamic changes in (A) heart rate, (B) mean arterial pressure, (C) carotid blood flow, and (D) brain oxygen saturation in sham-operated controls (open circle), and piglets resuscitated using CC+SI with a PIP of 10 $cmH_2O$ (CC+SI_PIP_10, closed circle), a PIP of 20 $cmH_2O$ (CC+SI_PIP_20, closed triangle), and a PIP of 30 $cmH_2O$ (CC+SI_PIP_30, closed square). Data are presented as mean (SD). * indicates a significant difference compared to sham-operated controls and its own baseline ($p<0.05$); # indicates a significant difference from baseline values ($p<0.05$); indicates a significant difference from sham group at the concurrent time point ($p<0.05$).

## Discussion

To optimize the effectiveness of CC+SI in newborn infants, the current study was performed to examine whether different PIP of the delivered SI would have an impact on ROSC, survival, and hemodynamic and respiratory outcomes in a newborn piglet model. To our knowledge, this is the first study investigating different PIP during CC+SI in resuscitation of asphyxiated piglets. The results of the study can be summarized as follows: i) resuscitation using CC+SI with pressures of 10, 20, and 30 $cmH_2O$ resulted in a similar time to ROSC; ii) CC+SI_PIP_20 had a trend to more survival compared to CC+SI_PIP_10 or CC+SI_PIP_30, iii) resuscitation using CC+SI with a pressure of 30 $cmH_2O$ delivered a larger $V_T$ and significantly lower exhaled $CO_2$; and iv) resuscitation using CC+SI with a pressure of 30 $cmH_2O$ resulted in significantly higher pro-inflammatory cytokine concentrations in the brain.

We have previously shown that using CC+SI with a PIP of 30 $cmH_2O$ can significantly improve ROSC and survival in asphyxiated newborn piglets, compared to using the

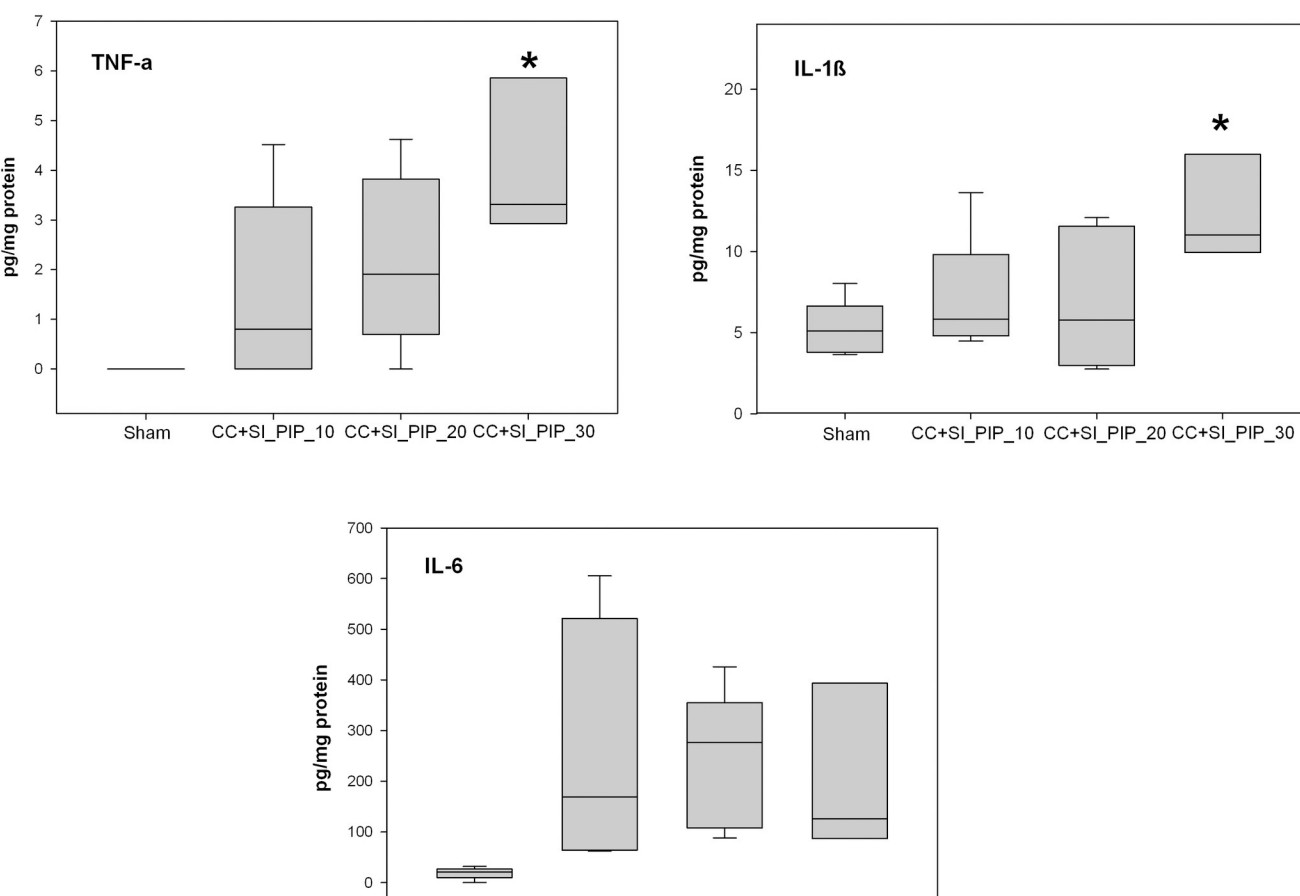

**Fig 3. Concentrations of pro-inflammatory cytokines TNF-α, IL-1β, and IL-6 in frontoparietal cortex tissue homogenates in sham-operated controls (n = 5), and piglets resuscitated using CC+SI with a PIP of 10 cmH₂O (CC+SI_PIP_10, n = 5), a PIP of 20 cmH₂O (CC+SI_PIP_20, n = 7), and a PIP of 30 cmH₂O (CC+SI_PIP_30, n = 3).** Results represent the median (solid bar), IQR (box margin), and 95% confidence interval. Asterisk (*) indicates a significant difference compared to sham-operated controls (p<0.05).

coordinated 3:1 compression-to-ventilation ratio (C:V) [20, 21, 23]. However, the optimal PIP for delivering SI during CC remains unknown. Current neonatal resuscitation guidelines recommend using a PIP of 20 to 25 cmH₂O for preterm infants and up to 30 to 40 cmH₂O in full-term infants as initial inflating pressure [12–14]. Although there is sufficient evidence that an inflation pressure of 20 to 25 cmH₂O delivers a high $V_T$ in preterm infants [15, 16], no study has ever measured the optimal PIP and the delivery of adequate $V_T$ during CC in the delivery room. Solevåg *et al* reported the need for a PIP of approximately 25 cmH₂O to achieve adequate $V_T$ in piglet cadavers and determined distending pressure during CC+SI [30]. This study suggests that chest recoil produces an inflation pressure-dependent $V_T$ allowing passive ventilation during CCs [30]. A similar result was reported while observing chest recoil after the application of a downward force on the chest of infants undergoing surgery requiring general anesthesia [31]. Additionally, the first randomized controlled trial comparing CC+SI and 3:1 C: V in preterm infants <33 weeks' gestation during neonatal resuscitation in the delivery room used a distending pressure of 24 cmH₂O reported adequate $V_T$ delivery [32]. All of the aforementioned studies suggest the optimal PIP for CC+SI is approximately 20–25 cmH₂O.

Notably, appropriate distending pressure is significant for proper ventilation of the neonate. While it is known that delivery of an adequate $V_T$ during CC is vital, it can be difficult to measure $V_T$ in the delivery room during 3:1 C:V using mask or endotracheal intubation [20, 27, 33]. Moreover, although several studies have demonstrated CC+SI can provide adequate $V_T$, relatively high distending pressure may cause higher $V_T$ [20, 33, 34]. Therefore, clinicians should be aware of the $V_T$ used during PPV and deliver <6–8 mL/kg, particularly in very preterm infants [19, 27, 33]. In this study, the delivered $V_T$ was significantly greater in piglets resuscitated with CC+SI_PIP_30, 14.0 (3.3) mL/kg, compared to the $V_T$ delivered when using CC+SI_PIP_10 and CC+SI_PIP_20: 7.3 (3.3) and 10.3 (3.1) mL/kg, respectively. High $V_T$ delivery during PPV (>8 mL/kg) has been shown to cause brain inflammation and injury in several animal studies [35–37]. Recently, Sobotka *et al* also reported that using a single SI for 30 seconds, followed by PPV in an asphyxiated near-term lamb model could result in the disruption of the blood brain barrier and cerebral vascular leakage [38]. These injuries may be caused either by a direct result of the initial SI or a higher $V_T$ delivered during subsequent ventilations [38].

Furthermore, a recent observational study in preterm infants <29 weeks' gestation observed a four-fold increase in intraventricular hemorrhage in infants ventilated even with a $V_T$ >6 mL/kg [19]. Overall, these studies showed the link between high $V_T$ delivery and the progression of brain injury. It is understood that high $V_T$ may cause brain inflammation and pathology through two mechanisms: hemodynamic instability, and a localized cerebral inflammatory response arising from ventilator-induced lung injury [39]. Interestingly, in our study, piglets resuscitated with CC+SI_PIP_20 showed lower mean arterial pressure, carotid blood flow, and $crSO_2$ while CC+SI_PIP_30 showed a lower carotid blood flow and $crSO_2$, when compared to baseline levels at the end of reoxygenation. This decrease in carotid blood flow for CC+SI_PIP_20 and CC+SI_PIP_30 may be due to alterations to pulmonary blood flow resulting from a high $V_T$, which have previously been shown in preterm lamb studies [37, 40]. This decrease in carotid blood flow may also cause hemodynamic instability, which can lead to brain injuries.

In addition, our study showed that piglets resuscitated with CC+SI_PIP_30 also exhibited higher levels of the pro-inflammatory cytokines IL-1β and TNF-α in frontoparietal cortex tissue. In a review by Polglase *et al*, it was suggested that excessive $V_T$ may initiate a pulmonary pro-inflammatory response and a systemic inflammatory cascade leading to brain injury [37]. This is also consistent with our findings, which suggested the large $V_T$ delivered by using a PIP of 30 $cmH_2O$ during CC+SI leads to the initiation of an inflammatory cascade, which in-turn increased pro-inflammatory cytokine levels in the brain. Brain injury may therefore be increased in the CC+SI_PIP_30 group by either of these two mechanisms. Although our use of 20 $cmH_2O$ PIP generated a $V_T$ >8 mL/kg, the lack of an inflammatory response in the brain tissue suggests it may be less injurious than a PIP of 30 $cmH_2O$. In terms of brain protection, a PIP of 10 $cmH_2O$ may appear to be the optimal pressure. Even though there were no statistically significant differences between groups, the hemodynamic recovery of the CC+SI_PIP_10 group tended to be better than both CC+SI_PIP_20 and CC+SI_PIP_30 groups during the four-hour observation period. However, in a clinical setting where the newborn is undergoing the fetal-to-neonatal transition, using a PIP of 10 $cmH_2O$ may not provide enough pressure to drive against any lung liquid retained within the airways. Further studies are needed to verify its effectiveness in the clinical setting.

Current neonatal resuscitation guidelines recommend 100% oxygen during neonatal CC [12–14], however the most effective oxygen concentration during CC remains controversial. To date, there are no available clinical studies regarding oxygen use during neonatal CC. Solevåg *et al* reported that asphyxiated piglets resuscitated with 21% vs. 100% oxygen have similar

time to ROSC (ranging from 75 to 592 s) with very high mortality rates (50–75%) in both groups during CC [41]. In addition, a recent meta-analysis of eight animal trials (n = 323 animals) comparing various oxygen concentrations during CC showed no difference in mortality rates and time to ROSC [42]. These results suggest that 21% oxygen has similar time to ROSC and mortality as 100% oxygen during CC. Therefore, we used 21% oxygen during CC in our experiment.

In the current study, the time to achieve ROSC for all groups was similar. Although there were more piglets that achieved ROSC following resuscitation with CC+SI_PIP_20, this was not significantly different compared to using CC+SI_PIP_10 and CC+SI_PIP_30. During CC +SI, improved carotid blood flow, mean arterial pressure, % change in ejection fraction, cardiac output, alveolar oxygen delivery and lung aeration may all result in faster ROSC by increasing intrathoracic pressure and improving minute ventilation [20–23, 32]. In our study there were significant differences in minute ventilation between groups, but there were no differences in carotid blood flow at 1 hour after ROSC. As there were no differences in carotid blood flow and the time to ROSC, carotid blood flow may be a major indicator in determining the time to ROSC.

In our study, the use of 30 $cmH_2O$ PIP resulted in lower exhaled $CO_2$ compared to 10 and 20 $cmH_2O$ (CC+SI_PIP_30: 10.8 (4.5) mmHg vs. CC+SI_PIP_10: 26.8 (8.5) mmHg vs. CC+-SI_PIP_20: 16.7 (10.6) mmHg; p = 0.0032). Li *et al* reported that exhaled $CO_2$, partial pressure of exhaled $CO_2$, and volume of expired $CO_2$, were significantly higher in surviving piglets compared to non-surviving piglets during resuscitation [33]. Chalak *et al* additionally reported a cut-off of 14 mmHg for exhaled $CO_2$ to be the most reliable indicator for ROSC with 92% sensitivity and 81% specificity [43]. The CC+SI_PIP_30 group had the lowest exhaled $CO_2$, which was below the 14 mmHg cut-off. Our results suggest that high $V_T$ and low exhaled $CO_2$ with a PIP 30 $cmH_2O$ may be associated with hemodynamic instability and increased pro-inflammatory cytokines, which may increase brain injury and decrease survival.

Our use of a piglet asphyxia model is a great strength of this translational study, as this model closely simulates delivery room events, with the gradual onset of severe asphyxia leading to bradycardia [23, 44]. However, several limitations should be considered before implementing CC during SI in the delivery room. Our asphyxia model uses piglets that have already undergone the fetal-to-neonatal transition, and piglets were sedated/anesthetized [21, 44]. Furthermore, our model requires piglets to be intubated with a tightly sealed endotracheal tube to prevent any endotracheal tube leak [23, 44]. Although endotracheal intubation may not occur in the delivery room as mask ventilation is frequently used, it is recommended when CC is administered [12–14]. The use of SI as an initial respiratory support technique might also be harmful, as indicated by a recent multicenter trial comparing SI versus PPV (SAIL [Sustained Aeration of Infant Lungs] trial) [45]. However, in our model SI was used during CC, rather than initial respiratory support as in the SAIL trial. Nevertheless, our findings are still clinically relevant as the distribution of cardiac output in the fetus and post-transitional neonate during asphyxia episodes are qualitatively similar.

## Conclusion

In asphyxiated term newborn piglets resuscitated by CC+SI, the use of different PIPs resulted in similar time to ROSC, but the use of a PIP 30 $cmH_2O$ showed a larger $V_T$ delivery, lower exhaled $CO_2$ and increased brain inflammation compared to using of a PIP 10 or 20 $cmH_2O$. Future studies in animal models and/or during neonatal resuscitation are needed to examine the optimal SI delivery parameters during CC.

## Author Contributions

**Conceptualization:** Po-Yin Cheung, Megan O'Reilly, Georg M. Schmölzer.

**Data curation:** Gyu-Hong Shim, Seung Yeun Kim, Po-Yin Cheung, Tze-Fun Lee, Megan O'Reilly, Georg M. Schmölzer.

**Formal analysis:** Tze-Fun Lee, Megan O'Reilly.

**Funding acquisition:** Georg M. Schmölzer.

**Investigation:** Gyu-Hong Shim, Seung Yeun Kim, Po-Yin Cheung, Georg M. Schmölzer.

**Methodology:** Gyu-Hong Shim, Seung Yeun Kim, Po-Yin Cheung, Tze-Fun Lee, Megan O'Reilly, Georg M. Schmölzer.

**Project administration:** Tze-Fun Lee, Georg M. Schmölzer.

**Resources:** Georg M. Schmölzer.

**Validation:** Georg M. Schmölzer.

**Writing – original draft:** Gyu-Hong Shim.

**Writing – review & editing:** Gyu-Hong Shim, Seung Yeun Kim, Po-Yin Cheung, Tze-Fun Lee, Megan O'Reilly, Georg M. Schmölzer.

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
