## [Decision Letter · Decision Letter 0]

4 Mar 2020

PONE-D-20-01119

Effects of sustained inflation pressure during neonatal cardiopulmonary resuscitation of asphyxiated piglets

PLOS ONE

Dear Dr. Schmölzer,

Thank you for submitting your manuscript to PLOS ONE. After careful consideration, we feel that it has merit but does not fully meet PLOS ONE’s publication criteria as it currently stands. Therefore, we invite you to submit a revised version of the manuscript that addresses the points raised during the review process.

We would appreciate receiving your revised manuscript by Apr 18 2020 11:59PM. To enhance the reproducibility of your results, we recommend that if applicable you deposit your laboratory protocols in protocols.io, where a protocol can be assigned its own identifier (DOI) such that it can be cited independently in the future. For instructions see: http://journals.plos.org/plosone/s/submission-guidelines#loc-laboratory-protocols

We look forward to receiving your revised manuscript.

Kind regards,

Jayasree Nair, MBBS MD FAAP

Academic Editor

PLOS ONE

Journal Requirements:

2. We ask that you please clarify in your methods section, whether the piglets were fully anesthetized during the 4hr recovery period. Please state whether any of the piglets died while not under anesthesia.

3. We noticed you have some minor occurrence of overlapping text with a previous publication,  https://doi.org/10.1016/j.resuscitation.2018.06.013, which needs to be addressed.

In your revision ensure you cite all your sources (including your own works), and quote or rephrase any duplicated text outside the methods section. Further consideration is dependent on these concerns being addressed.

Reviewers' comments:

Reviewer's Responses to Questions

**Comments to the Author**

1. Is the manuscript technically sound, and do the data support the conclusions?

Reviewer #1: Yes

Reviewer #2: Yes

2. Has the statistical analysis been performed appropriately and rigorously? 

Reviewer #1: Yes

Reviewer #2: Yes

3. Have the authors made all data underlying the findings in their manuscript fully available?

Reviewer #1: Yes

Reviewer #2: Yes

4. Is the manuscript presented in an intelligible fashion and written in standard English?

Reviewer #1: Yes

Reviewer #2: Yes

5. Review Comments to the Author

Reviewer #1: The authors present interesting and novel data on different inflation pressures during sustained inflation (SI) and uninterrupted chest compressions (CC) in an asphyxiated piglet model. The experiments were conducted at a reputable laboratory with extensive research experience on piglet models. Piglets were randomized into three groups: 1) SI of 10 cm H20, 2) SI of 20 cm H20 and 3) SI of 30 cm H20 and CC were given at a rate of 90 compressions/min. SI was provided for 20 s with 1 s rest intervals. First dose of epi (0.02 mg/kg) was given at 2 minutes (up to a total of 4 doses) and repeated every 3 minutes as needed if return of spontaneous circulation (ROSC) was not achieved. There was no difference in baseline characteristics between groups. Time and incidence of ROSC was not statistically significant, but in group 3 only 3/8 piglets achieved ROSC compared to 5/8 and 7/8 in groups 1 and 2, respectively. In addition, piglets in group 3 demonstrated higher concentrations of pro-inflammatory markers (TNF-alpha, Il-1Beta and IL-6) in frontoparietal cortex tissue homogenates.

This reviewer has the following remarks/suggestions:

Abstract, first sentence: This is a bold statement by the authors. The authors' claim that SI during CC reduces time to ROSC is supported by their previous observations in a piglet model. However, Vali et al did not show a difference in ROSC success or time to ROSC when comparing 3:1 compression-to-ventilation (C:V) to CC+SI in a asphyxiated cardiac arrest lamb model (PMID: 28661972). Even in their current study only 3/8 piglets achieved ROSC in the CC+SI PIP 30 cm H20 group.

In addition, in a pilot study comparing CC+SI to 3:1 C:V, 2/5 in the SI group died at <28d compared to 0/4 in the 3:1 C:V, which is a concerning trend.

There is lack of evidence to support this statement and the sentence should be revised.

Methodology: authors claim that the endotracheal tube was clamped until asystole, which they define as "no heart rate audible during auscultation and zero blood flow in the carotid artery."

However, in Figure 2, carotid blood flow is clearly not zero. The measured heart rate also does not reach zero.

Based on these values, the piglets are not in asystolic cardiac arrest as there is cardiac activity resulting in cardiac output (as demonstrated by left carotid blood flow).

The authors need to state that their model is one of severe bradycardia and not asystole.

Introduction, first paragraph, 3rd sentence: the referenced study (ref 9), which was published in 2007 collected data spanning the years 1991 to 2004 and is not recent. The authors should revise their sentence to more accurately reflect the period of the referenced paper.

Figure 2: are any values significantly different compared to baseline? The hatch (#) designation does not appear on the provided the figures.

Reviewer #2: Shim et al conduct an interesting experiment to determine the optimal peak inspiratory pressures to be used with a sustained inflation and chest compression. They demonstrated that a PIP of 30 centimeters results in larger TV but more inflammatory markers in the brain. They conclude that more studies are needed to determine the optimal SI delivery parameters during chest compression.

My major concern with this conclusion is that while NRP recommends intubation during CC most providers are giving mask PPV during the initial bradycardia. if a sustain inflation could restore circulation during chest compresssions it would be far more pragmatic to have a non-intubated model (or at least a comparator). I doubt providers are considering an SI breath wiht a PIP of 30 in humans. Most of the work will have to be done in animals since a human trial would be so difficult. I suggest the authors add a third arm to determine if the delivery of these breaths could be different with a face mask compared to an intubated animal.

6. PLOS authors have the option to publish the peer review history of their article (what does this mean?). If published, this will include your full peer review and any attached files.

Reviewer #1: No

Reviewer #2: No

---

## [Author Response · Author response to Decision Letter 0]

27 Apr 2020

Response to Reviewers

We would like to thank the editors and reviewers for their thoughtful comments. We have used them to improve the presentation of our manuscript.

Academic Editor

Response: We have revised the manuscript by referring to the PLOS ONE style template.

2. We ask that you please clarify in your methods section, whether the piglets were fully anesthetized during the 4hr recovery period. Please state whether any of the piglets died while not under anesthesia.

Response: We have edited the method section: “Throughout the entire experimental period” to indicate that animal was fully anesthetized during the 4h recovery period as well. We don’t have any data on whether any of piglets will die without anesthesia.

3. We noticed you have some minor occurrence of overlapping text with a previous publication, https://doi.org/10.1016/j.resuscitation.2018.06.013, which needs to be addressed.

In your revision ensure you cite all your sources (including your own works), and quote or rephrase any duplicated text outside the methods section. Further consideration is dependent on these concerns being addressed.

Response: This has been edited

Reviewer #1

1. Abstract, first sentence: This is a bold statement by the authors. The authors' claim that SI during CC reduces time to ROSC is supported by their previous observations in a piglet model. However, Vali et al did not show a difference in ROSC success or time to ROSC when comparing 3:1 compression-to-ventilation (C:V) to CC+SI in a asphyxiated cardiac arrest lamb model (PMID: 28661972). Even in their current study only 3/8 piglets achieved ROSC in the CC+SI PIP 30 cm H2O group. In addition, in a pilot study comparing CC+SI to 3:1 C:V, 2/5 in the SI group died at <28d compared to 0/4 in the 3:1 C:V, which is a concerning trend. There is lack of evidence to support this statement and the sentence should be revised.

Response: This has been edited.

2. Methodology: authors claim that the endotracheal tube was clamped until asystole, which they define as "no heart rate audible during auscultation and zero blood flow in the carotid artery." However, in Figure 2, carotid blood flow is clearly not zero. The measured heart rate also does not reach zero. Based on these values, the piglets are not in asystolic cardiac arrest as there is cardiac activity resulting in cardiac output (as demonstrated by left carotid blood flow). The authors need to state that their model is one of severe bradycardia and not asystole.

Response: To prevent bias for overall outcome, total cardiac arrest was determined by auscultation with a standard stethoscope by a single investigator (GMS), who was blinded to HR displayed by ECG and CBF. About 42% (10 out of 24) piglets still had low CBF (0 with IQR 0-9 mL); whereas the median HR was 13 (8~15)(with 38% [9 out of 24] showed pulseless electrical activity). There was a positive correlation between HR and CBF at cardiac arrest (r2=0.53, p=0.008). However, there were no correlations between HR and ROSC time (r2=0.35, p=0.098) or CBF and ROSC time (r2=0.18, p=0.393). The definition of asystole has been changed in the method section accordingly.

3. Introduction, first paragraph, 3rd sentence: the referenced study (ref 9), which was published in 2007 collected data spanning the years 1991 to 2004 and is not recent. The authors should revise their sentence to more accurately reflect the period of the referenced paper.

Response: This ha s been revised: A systemic review of newborns born between 1991 and 2004 who underwent prolonged chest compressions without signs of life at 10 minutes following birth noted 83% mortality, with 94% of survivors suffering death or severe disability [9].

4. Figure 2: are any values significantly different compared to baseline? The hatch (#) designation does not appear on the provided the figures.

Response: Figure 2 has been revised to add on statistical values as compared with its own baseline. Please note that the symbol designation has been changed to minimize numbers of symbol used for clarity.

Reviewer #2

1. My major concern with this conclusion is that while NRP recommends intubation during CC most providers are giving mask PPV during the initial bradycardia. if a sustain inflation could restore circulation during chest compressions it would be far more pragmatic to have a non-intubated model (or at least a comparator). I doubt providers are considering an SI breath with a PIP of 30 in humans. Most of the work will have to be done in animals since a human trial would be so difficult. I suggest the authors add a third arm to determine if the delivery of these breaths could be different with a face mask compared to an intubated animal.

Response: Thank you for this excellent suggestion: Unfortunately, our animal ethics committee has so far not allowed this approach, but given your comment we will resubmit an ethics application to compare intubation versus face mask during CC+SI.

---

## [Decision Letter · Decision Letter 1]

8 Jun 2020

Effects of sustained inflation pressure during neonatal cardiopulmonary resuscitation of asphyxiated piglets

PONE-D-20-01119R1

Dear Dr. Schmölzer,

We’re pleased to inform you that your manuscript has been judged scientifically suitable for publication and will be formally accepted for publication once it meets all outstanding technical requirements.

Kind regards,

Jayasree Nair, MBBS MD FAAP

Academic Editor

PLOS ONE

Additional Editor Comments (optional):

Reviewers' comments:

Reviewer's Responses to Questions

**Comments to the Author**

1. If the authors have adequately addressed your comments raised in a previous round of review and you feel that this manuscript is now acceptable for publication, you may indicate that here to bypass the “Comments to the Author” section, enter your conflict of interest statement in the “Confidential to Editor” section, and submit your "Accept" recommendation.

Reviewer #1: All comments have been addressed

Reviewer #2: All comments have been addressed

2. Is the manuscript technically sound, and do the data support the conclusions?

Reviewer #1: Yes

Reviewer #2: Yes

3. Has the statistical analysis been performed appropriately and rigorously? 

Reviewer #1: Yes

Reviewer #2: Yes

4. Have the authors made all data underlying the findings in their manuscript fully available?

Reviewer #1: Yes

Reviewer #2: Yes

5. Is the manuscript presented in an intelligible fashion and written in standard English?

Reviewer #1: Yes

Reviewer #2: Yes

6. Review Comments to the Author

Reviewer #1: (No Response)

Reviewer #2: my comments have been addressed. Thank you for performing this important work.

This will help support the ongoing work of the senior author in humans.

7. PLOS authors have the option to publish the peer review history of their article (what does this mean?). If published, this will include your full peer review and any attached files.

Reviewer #1: No

Reviewer #2: No

---

## [Editor Report · Acceptance letter]

11 Jun 2020

PONE-D-20-01119R1 

Effects of sustained inflation pressure during neonatal cardiopulmonary resuscitation of asphyxiated piglets 

Dear Dr. Schmölzer:

I'm pleased to inform you that your manuscript has been deemed suitable for publication in PLOS ONE. Congratulations! Your manuscript is now with our production department. 

Kind regards, 

on behalf of

Dr. Jayasree Nair 

Academic Editor

PLOS ONE